# Application of Bio-Active Elastin-like Polypeptide on Regulation of Human Mesenchymal Stem Cell Behavior

**DOI:** 10.3390/biomedicines10051151

**Published:** 2022-05-17

**Authors:** Vijaya Sarangthem, Harshita Sharma, Mohini Mendiratta, Ranjit Kumar Sahoo, Rang-Woon Park, Lalit Kumar, Thoudam Debraj Singh, Sujata Mohanty

**Affiliations:** 1Department of Biochemistry and Cell Biology, School of Medicine, Cell and Matrix Research Institute, Kyungpook National University, Daegu 41944, Korea; devi1703@gmail.com; 2Department of Pathology, All India Institute of Medical Sciences, New Delhi 110029, India; 3Stem Cell Facility, DBT Centre of Excellence for Stem Cell Research, All India Institute of Medical Sciences, New Delhi 110029, India; sharshita558@gmail.com; 4Department of Medical Oncology, All India Institute of Medical Sciences, New Delhi 110029, India; mohinimendiratta@gmail.com (M.M.); drranjitmd@gmail.com (R.K.S.); lalitaiims@yahoo.com (L.K.); 5Department of Medical Oncology Laboratory, All India Institute of Medical Sciences, New Delhi 110029, India

**Keywords:** elastin-like polypeptide, biopolymer, integrin, laminin, extracellular matrix, stem cells, cell differentiation

## Abstract

Regenerative medicine using stem cells offers promising strategies for treating a variety of degenerative diseases. Regulation of stem cell behavior and rejuvenate senescence are required for stem cells to be clinically effective. The extracellular matrix (ECM) components have a significant impact on the stem cell’s function and fate mimicking the local environment to maintain cells or generate a distinct phenotype. Here, human elastin-like polypeptide-based ECM-mimic biopolymer was designed by incorporating various cell-adhesion ligands, such as RGD and YIGSR. The significant effects of bioactive fusion ELPs named R-ELP, Y-ELP, and RY-ELP were analyzed for human bone-marrow-derived stem cell adhesion, proliferation, maintenance of stemness properties, and differentiation. Multivalent presentation of variable cell-adhesive ligands on RY-ELP polymers indeed promote efficient cell attachment and proliferation of human fibroblast cells dose-dependently. Similarly, surface modified with RY-ELP promoted strong mesenchymal stem cell (MSCs) attachment with greater focal adhesion (FA) complex formation at 6 h post-incubation. The rate of cell proliferation, migration, population doubling time, and collagen I deposition were significantly enhanced in the presence of RY-ELP compared with other fusion ELPs. Together, the expression of multipotent markers and differentiation capacity of MSCs remained unaffected, clearly demonstrating that stemness properties of MSCs were well preserved when cultured on a RY-ELP-modified surface. Hence, bioactive RY-ELP offers an anchorage support system and effectively induces stimulatory response to support stem cell proliferation.

## 1. Introduction

Mesenchymal stem cells (MSCs) are progenitor cells capable of self-renewal and differentiation to several mesenchymal lineages, including osteoblasts, adipocytes, and chondroblasts [1,2]. These physical characteristics are associated with organogenesis and tissue regeneration, which has attracted widespread interest from clinical researchers [3,4]. However, the restricted number of populations doubling and becoming senescent eventually greatly hindered their clinical application, as obtaining sufficient quantities of MSCs from culture was difficult [5,6]. The loss of stem cell characteristics after a long expansion in culture affects their immunomodulatory and differentiation capacities that limit their efficacy [7,8]. The regulation of stem cell behavior and survival were completely dependent on a complex microenvironment known as the stem cell niche. The extracellular matrix (ECM) is an essential component of the stem cell niche, which is involved directly or indirectly in regulation of proliferation, maintenance, self-renewal, and differentiation of stem cells [9,10,11,12,13]. Thus, natural and synthetic biomaterials mimicking ECM were being investigated for stem cells to grow and differentiate. The natural polymers, such as collagen, hyaluronic acid, fibrin, or alginate, or synthetic polymers, such as polyethylene glycol (PEG), dextran, or polyvinyl alcohol were commonly used [14,15]. However, due to the polydispersity, lack of cell interaction, possible cytotoxicity, and immunogenicity restricted the application of synthetic polymers [16]. Naturally derived polymers have inherent merits in this regard, including bioactivity, the ability to present receptor-binding ligands to cells, biodegradability, and susceptibility to natural remodeling, but antigenicity, instability, complexity of purification, and risk of disease transmission limit their application [17].

In this context, elastin-like polypeptide (ELP)-based biomaterials with enhanced biological and physical properties have opened the door to new approaches in stem cell applications. ELPs are a class of genetically encodable polymers that possess temperature-dependent phase transition properties [18]. The biocompatible, nontoxic, nonimmunogenic nature makes ELP ideal for various applications. ELP is composed of the pentapeptide sequence repeats [VPGXG]n (where X can represent any amino acid except proline and n indicates the number of repeating pentapeptides) originally derived from an amino acid motif found in the hydrophobic domain of human tropoelastin. Since ELPs are genetically encodable, it is relatively simple to fuse biologically active peptides or proteins to ELPs, creating ELP fusion proteins [19,20]. These ELP fusion proteins retain not only the inverse phase transition property of traditional ELPs, but also the biological activity of the fused peptide [21]. Hence, ELP fusion proteins can serve as an all-in-one matrix system, without the need for chemical conjugation. The stimulus-responsive phase transition inherent to ELPs has been used to purify engineered fusion proteins after protein expression in microbial hosts, leading to cost-effective production of functional chimeric proteins. ELPs have promising clinical application as systems for drug loading, targeting, and delivery in numerous fields, such as cancer therapy [22], regenerative medicine [23,24], or diagnostics [25]. Recently, we have demonstrated that ELP-based fusion proteins shield an appended biologically active peptide from proteolytic degradation by forming colloidal suspension of liquid-like coacervates [26], making this delivery system particularly well adapted for wound care treatment [27].

Typically, native extracellular matrices (ECMs) exhibit networks of molecular interactions between specific matrix proteins and other tissue components. ECM is a complex network of different macromolecules, such as fibrous proteins, including collagen, elastin, fibronectin, and laminin; proteoglycans (PGs); and glycosaminoglycans (GAGs), that confers well-defined physical, biochemical, and biomechanical properties to direct cell behaviors [28]. The interactions of stem cells with ECM can be directly activated by a number of cell receptors, including integrins, tegrins, syndecans, and other receptors. An increasing number of studies confirmed that integrins are core receptors involved in ECM–stem cell interactions and in the adhesion, anchorage, and homing of stem cells [29]. Accordingly, we have designed a matrix-derived protein chimera that contains RGD, the primary integrin-binding domain, and YIGSR, laminin-binding domain fused to an elastin-like polypeptide (ELP). This multifunctional ELP design offers a flexible protein engineering platform: (i) RGD, the primary integrin-binding domain of ECM proteins, mediates cell adhesion, proliferation, and migration; (ii) YIGSR binds to laminin receptors to activate signaling pathways and promote cell adhesion; (iii) ELPs, known to be well tolerated in vivo, provide a self-assembly scaffold with tunable physicochemical (viscoelastic, thermo responsive) properties. The concept used to design these fusion ELPs is a multivalent interaction of cells, based on integrin or laminin cell surface receptors, rather than monovalent interaction, in order to strengthen and stabilize ligand binding to the receptor and enhance cell proliferation and survival. It was aimed at stimulating biological responses by multiple interactions between ligands and receptors in order to regulate the function of diverse types of cells. Because ECMs play a key role in controlling stem cell behavior, it was expected that our fusion ELPs containing multi-ECM-binding domains would influence or enhance stem cell behavior to improve MSCs for clinical applications. Therefore, we sought to study the impact of multifunctional ELP variants on human fibroblasts and bone-marrow-derived stem cell attachment, early FA complex formation, cell proliferation, and maintenance of stemness properties of MSCs.

## 2. Materials and Methods

### 2.1. Culture of Bone-Marrow-Derived MSCs

Ethical clearance was obtained from Institutional Committee for Stem Cell Research (IC-SCR/110/20(R), All India Institute of Medical Sciences, New Delhi. All the methods described in this study were performed in accordance with the relevant guidelines and regulations of the Institution. Human bone marrow was harvested by iliac crest aspiration from healthy donors in the Department of Hematology. We have carefully optimized and standardized different isolation and expansion methods with or without gradient separation. The plating of whole bone marrow at a low cellular density has proven to be advantageous for mesenchymal stem cell (MSC) expansion. Isolation of MSCs via Ficoll density gradient centrifugation require approximately 20–50 mL of blood sample. Ficoll–Paque density gradient centrifugation (DGC) resulted in significant cell loss and influenced graft function. Since human blood sample availability was limited, we used a direct plating method of whole blood, for which only 5 ml of blood was required. We have strictly followed the International Society for Cellular Therapy (ISCT) proposed criteria to identify MSCs, such as: (1) the adherence to plastic; (2) the specific surface antigen expression (positivity for CD105, CD73, and CD90 and the lack of expression of CD45, CD34, CD14, or CD11b, CD79a or CD19 and HLA class II); (3) the multipotent capacity to differentiate into osteoblasts, adipocytes, and chondrocytes under standard in vitro differentiating conditions.

Briefly, for culturing MSCs, 1 mL of the bone marrow blood sample was seeded in each 60-mm culture plate and was maintained at 37 °C in a humidified atmosphere containing 5% CO_2._ Bone-marrow-derived stem cells (MSCs) were isolated and cultured in Dulbecco’s modified Eagle medium, low glucose (DMEM; Gibco, Grand Island, NY, USA), supplemented with 10% fetal bovine serum (FBS; Invitrogen, Carlsbad, CA, USA) and 1% antibiotics (10,000 IU/mL penicillin, 10,000 μg/mL streptomycin, and 25 μg/mL amphotericin B (Mediatech, Manassas, VA, USA)).

Green fluorescent protein (GFP)-tagged IHF (Human Fibroblast Cell Line) cells were grown in DMEM high glucose (Gibco, Thermo Fisher Scientific, Waltham, MA, USA) supplemented with 10% fetal bovine serum (Hyclone, Thermo Fisher Scientific, Waltham, MA, USA) and 100 U/mL penicillin and 100 mg/mL streptomycin (Sigma Aldrich, St. Louis, MO, USA). The media was changed every three days and the culture was maintained at 37 °C in a 95% humidified environment with 5% CO_2_. Cells were grown undisturbed until they reached 70% confluency.

### 2.2. ELPs Expression and Purification

BL21 (DE3) competent E. coli cells were further transformed with a modified pET 25b+ vector containing ELP, R-ELP, Y-ELP, and RY-ELP gene for protein expression. Starter cultures were prepared by inoculating transformed colonies in 15 mL of terrific broth media (IBI Scientific, Dubuque, IA, USA) containing 100 mg/mL ampicillin (Sigma Aldrich, St. Louis, MO, USA) for 6 h at 37 °C. Starter cultures were then transferred into 700 mL of fresh terrific broth media containing ampicillin and incubated for 16 h at 37 °C. The cells were then harvested by centrifugation at 4000 rpm for 20 min at 4 °C and suspended in a 15 mL lysis buffer. Cells were lysed by sonication at 4 °C, and ELP protein was purified using inverse transition cycling (ITC). Four rounds of ITC were performed to eliminate cell contaminants. All the protein samples were filtered with 0.2 µM syringe filters. The filtered protein samples were subjected to endotoxin removal by using Pierce High-Capacity Endotoxin Removal Spin Column (Thermo Scientific, Waltham, MA, USA) as per the manufacturers’ instructions. Endotoxin levels were measured using the Thermo Scientific Pierce LAL Chromogenic Endotoxin Quantitation Kit (A39552).

ELP purity was checked by SDS-PAGE, followed by Coomassie blue staining (Bio-Rad, Hercules, CA, USA). ELP concentration was measured by a UV-visible spectrophotometer (Agilent Technologies, Santa Clara, CA, USA) using an extinction coefficient of 5690 M^−1^ cm^−1^.

### 2.3. Cell Adhesion Assay

A 6-well plate was coated with 5 μM concentrations of fusion ELPs at 4 °C overnight. Human fibroblast cells (5 × 10^3^) were seeded on ELP-protein-coated plates and incubated for different time intervals (0–24 h) at 37 °C. Time-dependent cell adhesion was assessed by live imaging fluorescence microscopy based on GFP fluorescence (Etaluma, Lumascope 620, Carlsbad, CA, USA).

### 2.4. Phalloidin Staining

Phalloidin staining of MSCs was performed for clear visualization of cell adhesion and spreading at optimum ELP (5 μM) concentration. Briefly, coverslips were coated with respective fusion ELPs and adherence of cells was observed at different time points. At specific time intervals (6 h and 24 h), cells were fixed with 4% PFA for 15 min and washed extensively with PBS. Then, cell membranes were permeabilized with 0.5% Triton X-100 for 5 min. The cells were washed with PBS and stained with 50 μg/mL Alexa Fluor 488 phalloidin (Sigma Aldrich, St. Louis, MO, USA) for 40 min at room temperature in the dark. After 40 min, the unbound stain was washed with PBS and stained with Hoechst for 10 min at room temperature. Coverslips were mounted using DPX Mountant (Sigma Aldrich, St. Louis, MO, USA) and the cells were imaged using a fluorescence microscope (Leica, Wetzlar, Germany).

### 2.5. Cell Viability Assay

The 96-well plates were coated with different concentrations (0.625, 1.25, 2.5, and 5 μM) of ELP, R-ELP, Y-ELP, and RY-ELP at 4 °C overnight. Then, both IHF and MSCs, 3 × 10^3^ cells/well, were seeded on ELP-protein-coated plates and incubated for 48 h at 37 °C. The cell viability was assessed by CCK-8 kit. Briefly, after incubation of cells for a specific time interval, wells were washed with PBS, and then 10 μL of CCK-8 solution was added to each well. After 1–2 h of incubation, the absorbance of each well was quantified using the microplate reader at 450 nm.

### 2.6. Cell Proliferation Assay

To analyze the impact of fusion ELP on cell proliferation, 96-well plates were coated with 5 μM concentration of ELP, R-ELP, Y-ELP, and RY-ELP at 4 °C overnight. Both IHF and MSCs of 3 × 10^3^ cells/well were seeded on coated plates (5 μM) and incubated for different time intervals at 37 °C. Cell proliferation was assessed by CCK-8 kit at each time interval. After incubation, the absorbance of each well was quantified using the microplate reader at 450 nm after adding CCK-8 solution.

### 2.7. Population Doubling Assay

To examine the effect of fusion ELP coatings on the growth rate of MSCs, we measured the population doubling levels (PDLs) compared to the untreated control. A total of 0.6 × 10^4^ cells of MSCs (at P3) were seeded on 48-well plates. Upon reaching confluence, the cells were passaged at the same cell density. The population doubling time was determined by cumulative addition of total numbers generated after day 1, 3, 6, and 10. PDLs were calculated using the following formula (Equation (1)): PDL = log_2_ (collected cell number/seeded cell number)(1)

### 2.8. Migration Assay

A scratch assay was used to measure cell population migration on the influence of fusion ELPs. IHF and MSCs were cultured in 5% CO_2_ at 37 °C in DMEM high/low glucose media. Six-well polystyrene plates were coated with 5 μM ELPs at 4 °C overnight. After washing the plate with PBS, cells (2 × 10^5^ cells/well) were seeded and grown to confluence. 3 h prior to scratch assay, the culture medium was substituted with a fresh medium containing mitomycin (10 μg/mL; Sigma-Aldrich, St. Louis, MO, USA) to avoid the influence of cell proliferation. Then, the medium was removed and a cell-free gap was created by scratching the confluent cell layer with a sterile P10 micropipette tip. Cells were washed several times with PBS and replaced with fresh media. At specific time points, images of three nonoverlapping regions of each scratched zone were captured using a phase contrast microscope. Scratched wound recovery index (SWRI, %) was determined with the equation (Equation (2)): SWRI (%) = (total wound area − present wound area)/total wound area × 100(2)

### 2.9. Calcium Deposition Analysis

MSCs were seeded on a 5 μM ELP-coated 6-well plate and cultured to grow to 60% confluence. After 2 days, cells were cultured in DMEM medium containing 10% FBS supplemented with 10^−3^ M dexamethasone (Gibco, Thermo Fisher Scientific, Waltham, MA, USA), 50 μg/mL L-ascorbic acid 2-phosphate (Sigma-Aldrich, St. Louis, MO, USA), 10 mM β- glycerophosphate (Sigma-Aldrich, St. Louis, MO, USA), and 0.1 μM vitamin D (Enzo Life Sciences, Farmingdale, NY, USA) to induce osteogenesis. Medium changes occurred every 3 days, and the success of differentiation was assessed 7, 14, and 21 days after induction. Alizarin Red stain (Rowley Biochemical Institute, Danvers, MA, USA) was used to examine the calcium deposition on the ELP-coated plates 14 days after induction. The cells were rinsed twice with PBS, fixed with 60% isopropanol (Sigma-Aldrich, St. Louis, MO, USA) for 1 min, washed three times with ddH_2_O, and stained with 2% Alizarin Red stain for 5 min. Cells were then washed extensively with ddH_2_O and imaged using a bright field microscope (Nikon, Tokyo, Japan).

### 2.10. Lipid Droplet Staining

MSCs were seeded on respective ELP-coated 6-well plates and cultured to grow to 60% confluence. To induce adipogenesis, the culture medium was supplemented with 10^−6^ M dexamethasone, 0.5 mM 3-isobutyl-1-methylxanthine (IBMX), and 1 μg/mL insulin were purchased (Sigma-Aldrich, St. Louis, MO, USA). For adipocyte differentiation, lipid droplets were stained by Oil Red O staining on day 14. Briefly, post-induction cells were rinsed twice with PBS and fixed with 10% formalin (Thermo Fisher Scientific, Waltham, MA, USA) for 60 min at room temperature. The cells were then washed with PBS and incubated with 60% isopropanol for 5 min. Freshly prepared Oil Red O staining solution (4 mg/mL) was added to the cells and agitated for 20 min. After several washes with ddH_2_O to remove excess dye, cells were imaged using Nikon Bright Field Microscope. 

### 2.11. Alcian Blue Staining

The differentiation capacity of MSCs into chondrocytes was analyzed by culturing the cells (3 × 10^5^) in respective ELP-coated U-shaped 96-well plates in the presence of induction media. MSCs were cultured in high-glucose DMEM (Gibco, Thermo Fisher Scientific, Waltham, MA, USA), 1% antibiotics, 1% ITS+ Premix (6.25 μg/mL insulin, 6.25 μg/mL transferring, 6.25 μg/mL selenious acid, 1.25 mg/mL bovine serum albumin, and 5.35 μg/mL linoleic acid; BD Biosciences), 1 mM sodium pyruvate (Sigma-Aldrich, St. Louis, MO, USA), 50 μg/mL L-ascorbic acid 2-phosphate, 40 μg/mL L-proline (Fluka, St. Louis, MO, USA), 0.1 μM dexamethasone, 10 ng/mL TGFβ1 and 150 ng/mL BMP7 (Peprotech, Rocky Hill, CT, USA). Medium changes occurred every 3 days, and the success of differentiation was assessed 7, 14, and 21 days after induction.

Chondroblasts were fixed with 4% (*v/v*) formaldehyde for 20 min at room temperature. Afterward, cells were washed twice with ddH_2_O, followed by a 3-min incubation at RT with 3% (*v/v*) acetic acid (Merck Millipore, Germany, Cat. No. 100063) in ddH_2_O. Alcian Blue solution (0.1 g Alcian Blue 8GX (Sigma-Aldrich, St. Louis, MO, USA) diluted in 10 mL 3% (*v/v*) acetic acid in ddH2O at pH = 2.5) was added for 60 min, followed by a washing step with 1 M HCl for 3 min. Afterward, wells were washed four times with ddH_2_O and cells were imaged in Nikon bright-field microscope.

### 2.12. Western Blotting of Bone Marrow MSCs

Lysate of MSCs grown in ELP and ELP-fusion-coated plates were prepared using RIPA buffer. Cell lysates were centrifuged at 16,000× *g* value for 30 min at 4 °C and protein concentrations were determined using the BCA assay (Thermo Scientific, Rockford, IL, USA). Aliquots (30 mg) of total cellular proteins were separated by SDS-PAGE on 10–12% (*w/v*) gradient NuPAGE gels (Invitrogen), transferred to PVDF (Amersham Bioscience Buckinghamshire, UK), and incubated with specific antibodies. Immunoreactive proteins were detected by enhanced chemiluminescence.

Endogenous protein expression of type I collagen was assessed in MSCs grown in different ELP-coated plates. All the primary and secondary antibodies were purchased from Cell Signaling Technology (Danvers, MA, USA).

### 2.13. Analysis of Stemness Properties of MSCs

To study the effect of ELPs on MSC stemness properties, MSCs were cultured on ELP-coated plates with growth medium and were sub-cultured every 4–5 days. After the last passage (P10), cells were harvested and divided into three groups to perform the following assays: the expression level of hMSCs markers was investigated by flow cytometry. Harvested cells after the last passages were fixed in 10% formalin for 15 min at room temperature. After washing with PBS, cells were permeabilized with 0.1% triton X-100 for 15 min, followed by incubation in blocking buffer for 30 min and primary antibodies against CD45, CD73, CD90, and CD105 for 30 min or corresponding isotype control. After washing, cells were incubated with corresponding secondary antibodies for 30 min at RT. For each sample, 5000 events were counted and run on a flow cytometer (Merck Millipore, guava easy Cyte HT, Singapore). Lysate of MSCs grown in ELP- and ELP-fusion-coated plates were prepared using RIPA buffer. Western blot was performed after loading an equal amount of protein in SDS-PAGE to measure the endogenous protein expression of NANOG, OCT4, and SOX2 in MSCs grown in different ELP-coated plates. All the antibodies were purchased from Cell Signaling Technology.

### 2.14. Statistical Analysis

All data analysis was conducted using Graph Pad Prism 5 software (GraphPad Software, San Diego, CA, USA). Unless otherwise stated, *p* values were calculated using one-way ANOVA and error bars indicate the standard deviation of three separate experiments performed in triplicates. *** *p* < 0.0001, ** *p* < 0.001, and * *p* < 0.05 are considered statistically significant.

## 3. Results and Discussion

### 3.1. Genetic-Encoded Synthesis of Fusion ELPs and Protein Expression

Elastin, an elastomeric, insoluble, and fibrous protein, is one of the principal components of the extracellular matrix (ECM) and is mainly found in many supporting tissues in which high deformations are required (lung, skin, blood vessel, urinary bladder, and cardiac tissues). The recombinant methods of generation of elastin-based polymers by iterative recursive methods guarantees a high control of amino acid sequence that directs the synthesis of elastin biopolymers of high monodispertsity [18,30]. This exquisite control allowed the repetition of multiple copies of ECM-binding ligands (RGD and YIGSR) along the backbone of ELP with desirable properties and specific bio-functionalities for the target application in stem cell culture (Appendix A). The fusion ELP polymers, such as R-ELP, Y-ELP, and RY-ELP, of suitable length and molecular weight were synthesized using recursive directional ligation (RDL) method, as with our previous study [31]. Repeated rounds of RDL yielded polymers of different length, such as [R-V_6_]_n_, [Y-V_6_]_n_, and [RY-V_5_]_n_, where *n* = 2, 4, 8, and 16 repeats. The multifaceted polymers used in the study have the composition [R-V_6_]_16_, [Y-V_6_]_16_, and [RY-V_5_]_16_ containing 16 repeats of cell-adhesive domain (RGD and YIGSR) sequence. Later, the fusion ELPs were referred to as R-ELP, Y-ELP, and RY-ELP, respectively, in accordance with types of domain constitution. As schematically shown in Figure 1A, the fusion RY-ELP of desired length was ligated into pET 25 b+, which was already modified by insertion of gene cassette including SfiI enzyme site for ELP ligation. Proper ligation of ELPs in the vector was confirmed by double digestion with Nde I and Hind III, followed by DNA sequencing. Later, for ELP gene expression, pET 25 b+ plasmid harboring R-ELP, Y-ELP, and RY-ELP was transformed into BL21-D3 E. coli. Inherent thermal responsive phase transition characteristics of fusion ELP were exploited for purification by inverse transition cycling (ITC).

After removal of endotoxin from the protein samples, the endotoxin levels of the ELP, R-ELP, Y-ELP, and RY-ELP were found to be 0.215, 0.211, 0.217, and 0.206 EU/mL, respectively. The expression levels and purity of ELPs were confirmed by SDS-PAGE, followed by Coomassie blue staining. Gel electrophoresis analysis revealed a single band with the appropriated molecular weight of ~47.9 kDa for R-ELP, ~49.3 kDa for Y-ELP, and ~50.5 kDa for RY-ELP, confirming the efficiency of the process to purify the polypeptides from bacterial lysate (Figure 1B). The sharp single bands of respective ELPs approximately corresponding to theoretical calculated molecular weight based on amino acids demonstrated the possibility of successful removal of impurities after four-round ITC. In contrast to other polymers, these fusion ELPs can be purified cost-effectively. It provided an approach to eliminate the expensive mode of purification by chromatography and replace it with simplified large-scale purification using the ITC method with relatively high yields (~100–400 mg/L).

### 3.2. Thermal Characteristics of Fusion-ELPs

The thermal responsive sol-gel transition of ELPs is a useful attribute for utilization in regenerative medicine and tissue engineering purposes. We further investigated the transition temperature, Tt, of respective fusion ELP proteins at the range of 20–55 °C in 1 °C/min increment (Appendix A). The entropically driven phase transition in response to increasing temperature was evaluated by turbidimetry at 25 µM concentration. Temperature-dependent phase separation was observed for all biopolymers. The Tt was found to be 34.71 °C, 21.5 °C, and 22.21 °C for R-ELP, Y-ELP, and RY-ELP, respectively. For a similar concentration of 25 μM, Y-ELP and RY-ELP displayed depressed Tt up to 14 or 13.3 °C compared to control ELP (Tt 35.5 °C). The negative correlation between Tt and the fraction of hydrophobic surface area on ELP-fused proteins is well established [32]. When YIGSR was incorporated repeatedly in either Y-ELP or RY-ELP, the transition temperature of ELP decreased, whereas substituting RGD did not substantially change the ELP transition temperature. All fusion ELPs form insoluble aggregates above Tt and exhibit sol-gel transition. Polymer suspension progressively dissolved when temperature decreased under 20 °C. Incubation below 15 °C led to a complete loss of turbidity after thermally triggered aggregation, demonstrating that the phase transition is reversible. A thermo-induced sol-to-gel transition will likely lead to the formation of coacervate layers or thin films, which can be used to embed ELPs on bioinert surfaces that can support the morphological characteristics and functions of different types of cells.

### 3.3. Effect of Fusion ELP on Cell Adhesion and Proliferation

Surface modification of elastin-like polypeptides (ELP) with peptides derived from ECM proteins gained much interest in the tissue engineering and regenerative medicine fields [33]. It offers a versatile way to modulate interactions between cells and the ECM matrix components critical for inducing cellular adhesion, spreading, and migration. Thus, designed ELP derivatives were further examined for their role in enhancing adhesion and cell survival. Mesenchymal stem cells (MSCs), derived from different tissues, are characterized by a fibroblast-like morphology. Many studies have revealed that MSCs share many characteristics with fibroblasts, such as being present in almost every human tissue, playing vital roles in wound healing, and representing a heterogeneous population of cells [34]. Fibroblasts were evidenced to express many of the same markers as MSCs, in that they can be induced to multilineage differentiation similar to MSCs [35,36]. It was also reported to meet all the standards set by the International Society for Cellular Therapy (ISCT) for characterization of MSCs; thus, the effect of fusion ELPs on cell attachment and proliferation was primarily examined in human dermal fibroblasts [37,38,39,40,41]. All ELPs containing RGD and YIGSR or both were estimated for their association with fibroblasts, a key constituent of the dermal layer. The ability of ELPs to bind IHF cells was evaluated using a crystal violet assay (Appendix A). When IHF was allowed to attach at a fusion-ELP-coated surface, the percentage of cell attachment was significantly increased in a dose- and time-dependent manner (Appendix A). RY-ELP showed greater cell adhesion of IHF compared to other fusion ELPs. Simultaneously, time-dependent cell adhesion was assessed by live imaging using GFP stably expressed IHF cells. It was further confirmed that Y-ELP and RY-ELP displayed faster cell adhesion and proliferation compared with uncoated or ELP controls (Figure 2A). The significant adhesion was also observed in R-ELP-coated plates as well. The engagement of multiple copies of both domains on ELP, indeed, provide a firm cell adherence matrix for IHF.

Several ECMs contain multiple protein-binding domains, such as RGD or YIGSR sequences, and play a critical role in the proliferation of various types of cells. The viability of IHF cells on ELP-, R-ELP-, Y-ELP-, or RY-ELP-coated surfaces was studied for different time intervals using CCK-8 assay. In all R-ELP-, Y-ELP-, or RY-ELP-coated surfaces, the percentage of cell viability increased dose-dependently. Among them, RY-ELP showed increased cell viability and proliferation in comparison to other groups (Figure 2B,C). Overall, ELP control had the least effect on cell attachment or cell proliferation, which directed that ELP did not possess any stimulatory effect but maintained viability of the cells. The conjuncture of both ECM domains showed a synergistic effect by facilitating faster IHF attachment, which, in turn, provides structural support needed for cell survival and proliferation.

### 3.4. Induction of Migratory Effect by Fusion ELP on IHF

The efficient motility of endogenous cells from wound margins into the injured site is essential for the wound healing process. To assess the stimulatory effect of ELP derivatives on IHF cell migration, a scratch assay was performed (Figure 3A). All fusion ELPs substantially accelerated the migration of IHF cells in a time-dependent manner. The highest rate of migration was recorded by RY-ELP at 24 h compared to another group. At a 5 μM coating concentration of ELP control, stimulatory effect on IHF motility was rarely observed. The accelerated motility of IHF was observed on R-ELP- and Y-ELP-coated plates as well. Estimation of migratory effect through scratched wound recovery index (SWRI, %) revealed 91.8 and 98.9% by Y-ELP and RY-ELP after 24 h, while ELP and R-ELP displayed 47.4 or 69.4% of the recovery index (Figure 3B). Simultaneously, the measurement of migratory cells per field revealed 2.85- or 2.34-fold increase in RY-ELP- or Y-ELP-coated groups, respectively (Figure 3C), whereas R-ELP displayed a 1.82-fold increase in migratory cells and no significant changes were detected in control ELP compared with uncoated surfaces. Accordingly, we confirmed that our designed RY-ELP could recapitulate the biophysical properties of the ECM, enhancing migration. It was anticipated that RY-ELP would induce a stimulating effect on IHF cells, which, in turn, activated signaling cascades that enhanced cellular migration.

### 3.5. Interaction of MSCs on ELP Matrix

For wound healing, stem cell therapy is one of the emerging treatment modalities with high potential for restoring damaged tissue. RGD and YIGSR are integrin- and laminin-binding ligands of ECM proteins, and they are extensively used to promote cell attachment, including stem cells [42]. Thus, fusion ELPs were further examined for their effect on cellular adhesion of human bone-marrow-derived mesenchymal stem cells. To understand the ability of fusion ELP on the attachment, the maintenance of the cells and strength of MSC attachment on a modified surface was investigated and quantified. The adhesion capabilities of MSCs on fusion ELP-modified surfaces at various time intervals were confirmed by phalloidin staining. The incorporation of ECM domains, such as integrins and laminin-binding ligands (RGD and YIGSR), triggered greater FA complex formation as a cell-contact point, thereby improving the cell-adhesion process. It was found that R-ELP-, Y-ELP-, and RY-ELP-coated surfaces stimulated profound formation of many filopodia-like extrusions and cell-contact points (Figure 4A,B). However, RY-ELP displayed a superior cell-adhesion process at early incubation time by increasing FA complex formation (Figure 4A). Together, the cells cultured on an RY-ELP-coated surface had a well-attachment morphology, with elongated extrusions and the shape of a well-attachment cell (Figure 4B). Both R-ELP and Y-ELP showed similar influences on FA formation and cell attachment. The engagement of these two functional motifs may cause a synergistic effect on signaling proteins related to the FA complex cell adhesion process, which, in turn, regulates proper attachment and cell survival processes. This study evidenced that all recombinant fusion ELPs markedly increased MSC adhesion. The fusion ELPs well mimicked the topology of ECM matrix and provided a matrix for adhesion-related cell functions and stability to MSCs.

Additionally, it has been shown that integrins cluster upon association with ECM ligands, leading to the formation of focal adhesions that regulate force transmission [43,44]. Integrin-mediated cell adhesion depends on a proper spacing to form a focal adhesion [45]. The strategy of multiple constitution of ECM-binding ligands on an ELP backbone have a significant impact on focal adhesion formation and stimulating the cellular response. It was clearly validated that spacing between the ligands in the designed polymers did not cause any steric hindrance, but defined spatial arrangements lead to optimal cellular activity. 

### 3.6. Effect of Fusion ELP on Viability and Proliferation of MSCs

Biomaterial-based techniques that replicate stem cell niches are important for preserving stem cell behavior and promoting stem cell survival. Thus, fusion ELPs were investigated for their role in supporting MSC proliferation. Fusion ELPs, such as R-ELP-, Y-ELP-, and RY-ELP-coated plates, showed enhanced cell adhesion compared with the uncoated control (Figure 5A and S3). The best condition for cell adhesion was observed in RY-ELP-coated surfaces. Estimation of adhered cells displayed 1.49-, 3.03-, and 3.81-fold increase in R-ELP, Y-ELP, and RY-ELP compared with uncoated surfaces. The surface chemistry affected the number of adherent cells. Simultaneously, the viability of MSCs on ELP-, R-ELP-, Y-ELP-, or RY-ELP-coated surfaces monitored after 24 h revealed an increase in the percentage of cell viability in a dose-dependent manner. Among them, RY-ELP showed an increased cell index ratio in comparison to other groups (Figure 5B). Further, evaluation of cell proliferation rate at different time intervals showed RY-ELP-coated surface induced greater MSC proliferation compared with the control (Figure 5C). ELP control showed lower cell index ratio, indicating that ELP did not play a crucial role in cell attachment but acted as a polymer backbone to support the presentation of ECM ligands in proper conformation to form a cell-adhesive matrix.

As a result of their short shelf life and accelerated senescence, MSCs are not readily accessible for clinical applications. Following validation of the morphology of adherent cells and viability of MSCs cultivated on fusion-ELP-coated surfaces, the inductive properties of the ELPs in growth rate were optimized. MSCs were cultured in a fusion-ELP-modified surface and, upon confluency of the cells, PDTs were calculated. Cells cultured for 7 days showed a PDT similar to that of the short-term, 2 days. In contrast, cells cultured for more than 7 days showed a significantly higher PDT, indicating that it took those cells much longer to form a confluent monolayer (Figure 5D). In accordance with the growth curve, the population doubling times of R-ELP, Y-ELP, and RY-ELP were 13.3 ± 0.52, 13.3 ± 0.91, and 13.3 ± 0.53, respectively, while untreated and ELP controls have a similar population doubling time of 25.4 ± 0.31 h. Hence, R-ELP, Y-ELP, and RY-ELP were predicted to lessen PDT of MSCs up to ~2-fold compared to controls, thus confirming the enhanced cellular proliferation. Additionally, it appeared that there were no significant changes in the rate of proliferation post 7 days in untreated or ELP groups, whereas cells showed remarkable increases in growth rate, even after 10 days in the case of Y-ELP and RY-ELP groups (Figure 5D). Collectively, ECM mimetic fusion ELP is attributed to provide structural support and form an environment favorable to stem cell survival and proliferation, which are crucial for tissue regeneration. This multivalent pattern of adhesive epitopes at distinct surface densities had significant benefits in stimulating the growth of MSCs.

### 3.7. Effect of Fusion ELP on Migratory Effect of MSCs

To assess the effects of fusion ELPs on the behavior of MSCs, including proliferation and migration, a scratch assay was performed (Figure 6A). All fusion ELPs substantially accelerated the migration of MSCs in a time-dependent manner. At a 5 μM coating concentration of ELP control, the stimulatory effect on MSC motility was rarely observed, as with the untreated control; however, the highest rate of migration was recorded in the case of RY-ELP at 24 h. Likewise, the accelerated motility of MSCs was observed on R-ELP- and Y-ELP-coated plates, but lesser in comparison with RY-ELP. The effect of different fusion ELPs on the migration of MSCs was evaluated by measuring the scratch wound recovery index (SWRI) in correspondence to reduction in cell-free areas mimicking the wound healing process. After 24 h, except for the control group, all experimental groups exhibited cell migration, with a significant decrease in cell-free areas (Figure 6B,C) (*p* < 0.001). The analysis of the scratched wound recovery index (SWRI, %) revealed wound recovery of 63.7 and 81.5% by Y-ELP and RY-ELP after 24 h, while ELP and R-ELP displayed a minimum recovery index of 18.1 or 23.8% compared to the control (15.9%). After 48 h, cell migration was recorded in all groups, including the control group, through a significant decline in cell-free areas compared with those at the 24 h time point. Hence, the propensity of cell migration is similar in all groups, which means that cells in all groups tend to fill the wounded areas after being scratched. Nonetheless, each group affected cell migration at distinct levels. At 24 h time points, both Y-ELP and RY-ELP had the narrowest cell-free areas and significantly lower levels than all other groups (*p* < 0.001), although RY-ELP showed the lowest cell-free areas, which were significantly narrower than another group (*p* < 0.001). Correspondingly, the number of migratory cells in the scratch area was predominant in the RY-ELP-coated surface than other groups. Accordingly, it could be interpreted that RY-ELP stimulated MSC migration with the best outcomes, even better than standard cell culture medium. At the same time, measurement of migratory cells per field revealed 12.0- or 9.3-fold increase in RY-ELP- or Y-ELP-coated groups, respectively (Figure 6C). However, R-ELP displayed a 2.4-fold increase in migratory cells and no significant changes were detected in control ELP in comparison with uncoated surfaces. Thus, the presence of RGD and YIGSR on ELP has a great impact on stimulating the MSCs’ migration.

### 3.8. Effect of Fusion ELP on ECM Secretion

In recent years, there has been immense attention to MSCs, as a primary candidate for cell-based therapies. The great potential of MSCs has been attributed to the array of MSC secretome, secretion of which is essential for tissue regeneration, apart from their differentiability, antiapoptotic, anti-inflammatory roles, immune suppression, and migration abilities to the injured tissues [46]. Secretome includes soluble paracrine factors, such as proangiogenic, antiapoptotic, immunomodulatory, anti-scarring, chemo-attractive, and mitogenic, required for tissue regeneration [47]. In addition, MSCs also secrete various ECM proteins, proteases, and their inhibitors, which play a crucial role in matrix rebuilding and remodeling [48]. Thus, the presence of type I collagen on fusion-ELP-modified surfaces was qualitatively visualized from immunofluorescence images compared to controls at various time points (Figure 7A,B). The images of the cells in respective groups were obtained after normalization with fluorescence intensity of uncoated surface. All fusion-ELP-coated groups displayed detectable type I collagen intensity at day 10, with none detectable in day 1 cultures from images. The intensity of collagen I was significantly higher in Y-ELP and RY-ELP than other groups (Figure 7B). Furthermore, Western blot analysis of MSCs grown on ELP-, R-ELP-, Y-ELP-, and RY-ELP-coated plates showed that Y-ELP- and RY-ELP-coated plates showed significant increases in the expression of endogenous type I collagen protein when compared to control and ELP-coated group (Figure 7C).

Thus, the presence of RGD and YIGSR peptide on ELP backbone promotes significant deposition of type I collagen protein compared to control. The greater induction of collagen deposition was highly anticipated to influence MSC survival, proliferation, migration, and other functional responses.

### 3.9. Effect of Fusion ELP on Stemness Property of MSCs

It has been reported that the interaction between ECM with integrin receptors affect embryonic stem cell (ESC) differentiation [49]. The impact of fusion ELP on the stemness property of MSCs was determined by analyzing the expression of the cell surface markers CD73, CD90, and CD105. The flow cytometry analysis of the cells cultured on fusion-ELP-coated dishes revealed normal expression of standard hMSC markers (Figure 8A–C). The expression of positive markers CD73, CD90, and CD105 was similar in all fusion ELPs parallel with untreated control (Figure 8D and Appendix A). Endogenous protein expression of Nanog, SOX2, and OCT4 was almost similar in all the coated plates compared to control (Figure 8E). These results verified that fusion ELPs promoted stemness in a manner similar to the stem cell niche. Taking these results together with those from the cell proliferation assay, all fusion ELPs effectively maintained multipotent abilities in MSCs. Thus, decoration of ECM-binding ligands on ELP-favored stem cells was found to hold the multilineage potential.

### 3.10. Effect of Fusion ELP on Trilineage Differentiation of MSCs

Many studies have indicated that proliferation and multilineage potential are fundamental progenitor properties of MSCs [50]. To harness the robust therapeutic potential of MSCs, the ability to preserve MSC progenitor potency is important, which is often abrogated by extensive monolayer culture. Therefore, we investigated whether designed fusion ELPs could provide a matrix with the features resembling native stem niche and preserve the progenitor potency. The effect of fusion ELPs on trilineage differentiation of MSCs was determined after plating the cell on fusion ELP surfaces. To evaluate the influence of fusion ELP on the trilineage differentiation ability of MSCs, differentiation to adipocytes, osteocytes, and chondrocytes was achieved according to the manufacturer’s protocol. After induction, MSCs were stained with Alizarin Red, Oil Red O, and Alcian Blue to observe osteogenesis, adipogenesis, and chondrogenesis, respectively. It was observed that MSCs were well differentiated to adipocytes, osteoblasts, and chondroblasts in respective in vitro tissue culture differentiating conditions (Appendix A). Interestingly, a significant increase in the differentiated cell population clearly demonstrated that MSC differentiation capacity was well preserved on fusion-ELP-coated surfaces.

Lipid droplets were visualized by Oil Red O staining and showed the presence of lipid droplets after 21 days of maintenance in adipogenic medium (Figure 9A). It was found that Y-ELP- and RY-ELP-coated surfaces had higher Oil Red O staining than R-ELP- or ELP-coated groups. Taken together, these data demonstrated that increased interaction of YIGSR residue with the laminin receptor of MSCs probably induced lipid droplet accumulation.

The differentiation to osteoblasts was confirmed by staining with Alizarin Red to examine calcium nodule formation. The Alizarin Red stain indicated the presence of calcium nodules on all groups, but calcium deposition was abundant on RGD-ELP-, Y-ELP-, and RY-ELP-coated surface compared to uncoated or ELP-coated surface (Figure 9B). Interestingly, enhanced calcium nodule formation on RGD or YIGSR containing ELP (R-ELP, Y-ELP, and RY-ELP) coated groups was far greater than that of noncoated surface. This result clearly evidenced the importance of RGD or YIGSR and their interaction with integrin or laminin receptors on induction of matrix mineralization.

Chondrogenic differentiation of MSCs on respective fusion-ELP-coated surfaces resulted in the formation of cartilage with its typical extracellular matrix post 14-day induction with chondrogenic differentiation medium. Further verification of cartilage formation by cytochemistry staining with Alcian Blue revealed all the fusion ELPs formed intense blue-colored spheroid with condensed cartilage formation (Figure 9C). Amongst these, RY-ELP-coated surface formed bigger spheroids with intensely blue colored cartilage extracellular matrix compared to other groups. Both uncoated and ELP groups did not show any considerable spheroid formation.

The protein composition of ECM is variable, tissue-specific, and provides essential scaffold and biochemical signals required for specialized cell growth, tissue homeostasis, and development. Y-ELP and RY-ELP have significant effects on the adipogenic differentiation (Appendix A), whereas R-ELP, Y-ELP, and RY-ELP have a greater effect on osteogenic differentiation. This clearly indicated that YIGSR peptide might be responsible for signaling involved in adipogenic differentiation, whereas both RGD and YIGSR for osteogenic differentiation. The synergistic effect of both the ECM ligands was observed in the case RY-ELP in chondrogenic differentiation. Thus, our designed biopolymers have remarkable influence on trilineage differentiation and tissue specificity based on ECM ligand constitution.

## 4. Conclusions

The genetically engineered cell-adhesive, thermal-responsive elastin-like polypeptides showed a great potential in maintaining stem cell behavior. The designed fusion ELPs represent a humanized inert biomaterial with their bioactivity, biocompatibility, and intrinsic structure resembling the native ECM matrix. Various in vitro studies have demonstrated that fusion ELPs provide support and anchorage for MSCs, thereby enhancing adhesion, survival, proliferation, migration, and differentiation. The incorporation of ECM ligands (YIGSR and RGD) was shown to have synergistic positive effects on cell adhesion, proliferation, and migration of MSCs cells. The RY-ELP matrix evidently provided a better support and inducible microenvironment for the stem cells, mimicking the complex natural ECM. Additionally, all fusion ELPs have significant effects on the adipogenic, osteogenic, and chondrogenic differentiation in a tissue-specific manner. So, further incorporation of stem cells into well-structured 3D scaffolds will increase the competence of restoring and repairing dysfunctional tissues. Further modification of fusion ELPs for bio-fabrication enables the precise design and scalable biomanufacturing of cell-based scaffolds using program-controlled bioprinting or bio-assembly for potential use for clinical application.

## Figures and Tables

**Figure 1 biomedicines-10-01151-f001:**
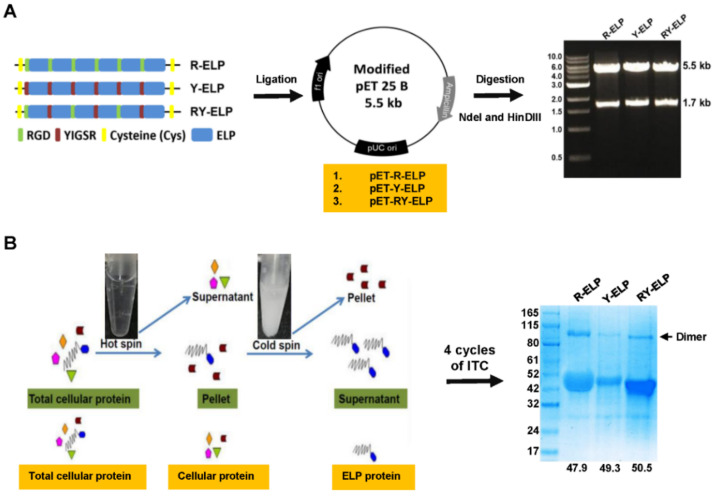
Expression and purification of ELP derivatives. (**A**) Schematical representation of R-ELP, Y-ELP, and RY-ELP ligation into pET 25 b+ after linearization with SfiI enzyme. Similarly, all the fusion ELP (R-ELP, Y-ELP) was ligated into pET 25 b+ vector for protein expression. Proper ligation of fusion ELPs in the vector was confirmed by double digestion with Nde I and Hind III (right). Size of the DNA fragments after digestion was indicated on the right. (**B**) All the fusion ELPs were purified by triggering phase transition at room temperature by adding NaCl. Repeated cycle of hot spin to aggregated protein and cold spin to dissolve ELP pellet was continued for four rounds to obtain fusion ELPs from cell lysate to homogeneity. The purity of fusion ELPs were confirmed through SDS-PAGE analysis (on right). The molecular weight fusion ELPs were indicated below (KDa).

**Figure 2 biomedicines-10-01151-f002:**
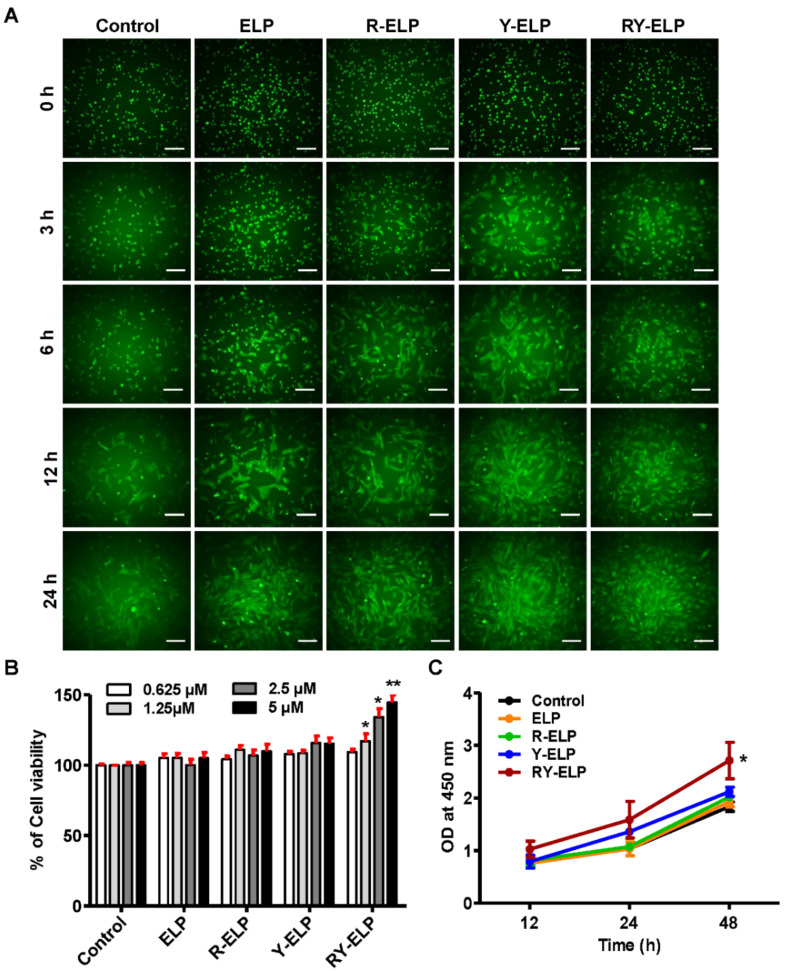
Cell adhesion, viability, and proliferation assay of IHF cells. (**A**) Live imaging of IHF cell attachment on fusion ELP (ELP, R-ELP, Y-ELP, and RY-ELP) modified surface at different time intervals (*n* = 3), Scale bars: 50 μm. (**B**) Cell viability of IHF cells was evaluated post 24 h cultured on a fusion-ELP-coated surface (*n* = 3), ** *p* < 0.01 and * *p* < 0.05 significant difference of respective fusion ELPs compared with uncoated control. (**C**) The proliferation rate of IHF cells on fusion-ELP-modified surfaces was evaluated at different time intervals (12, 24, and 48 h) (*n* = 3), * *p* < 0.05 significant difference for RY-ELP compared with uncoated surface (control).

**Figure 3 biomedicines-10-01151-f003:**
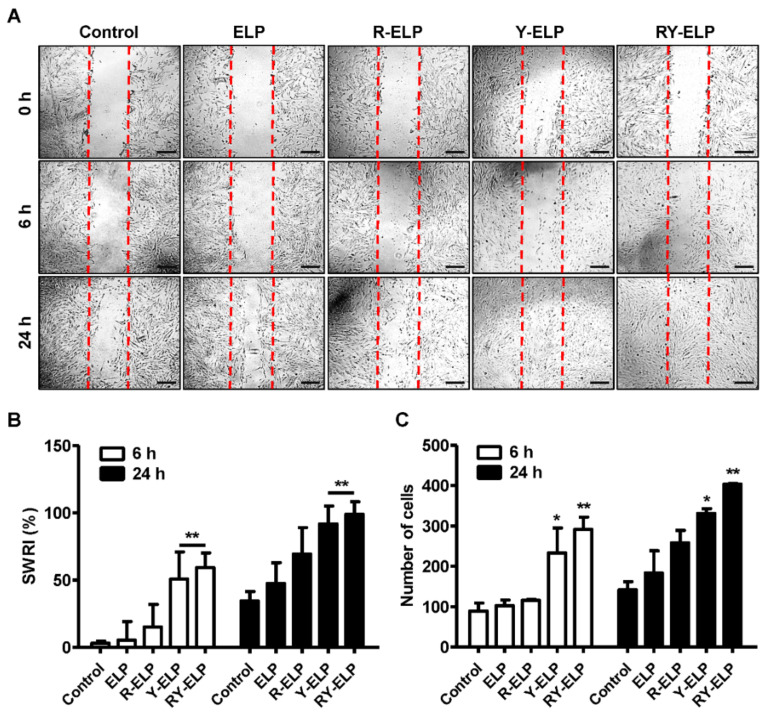
Migration assay of IHF cells. (**A**) Representative phase-contrast images of wounded IHF cells cultured on a fusion-ELP-coated surface as compared with uncoated over a 24-h period (*n* = 3), Scale bars: 100µm. (**B**) The bar graph represents the percentage of scratch wound recovery index (SWRI). ** *p* < 0.01 significant difference for Y-ELP and RY-ELP compared with untreated control (*n* = 3). (**C**) The number of migratory IHF cells at different time points (*n* = 3), * *p* < 0.05 and ** *p* < 0.01 significant difference of Y-ELP or RY-ELP compared with uncoated control.

**Figure 4 biomedicines-10-01151-f004:**
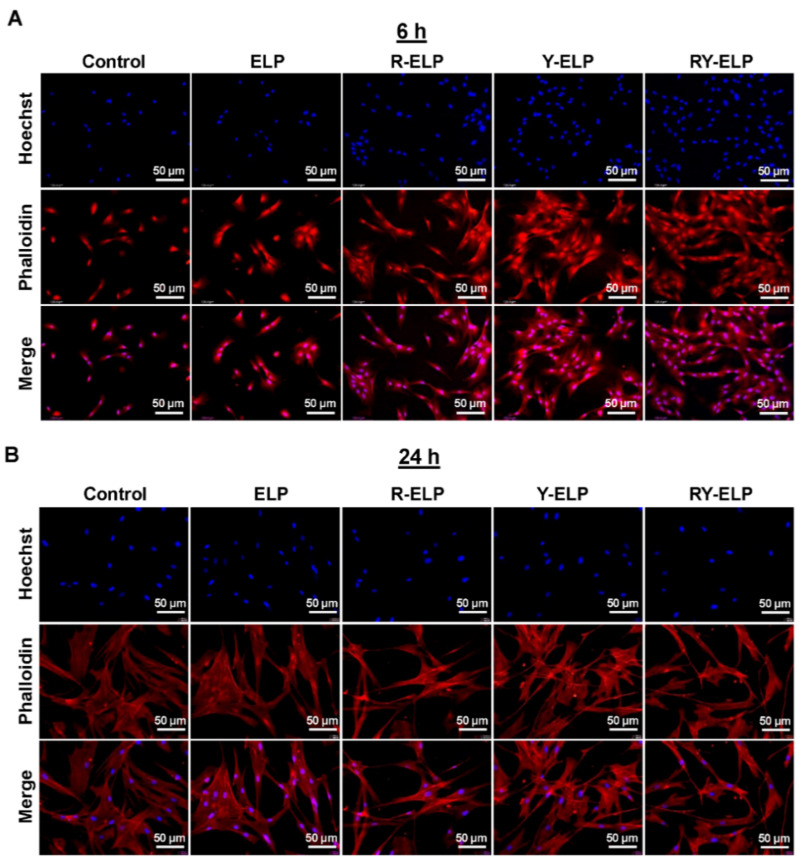
Determination of focal adhesion formation by Phalloidin staining. MSCs was seeded on fusion-ELP-coated coverslips for various time intervals and the adhered cells were fixed and labeled for nuclei (blue) by Hoechst 33,258 and actin filament (red) phalloidin–TRITC, respectively. Representative fluorescence microscopy images of MSCs adhering on the fusion-ELP-coated surface at 6 h (**A**) and 24 h (**B**), Scale bars: 50 μm.

**Figure 5 biomedicines-10-01151-f005:**
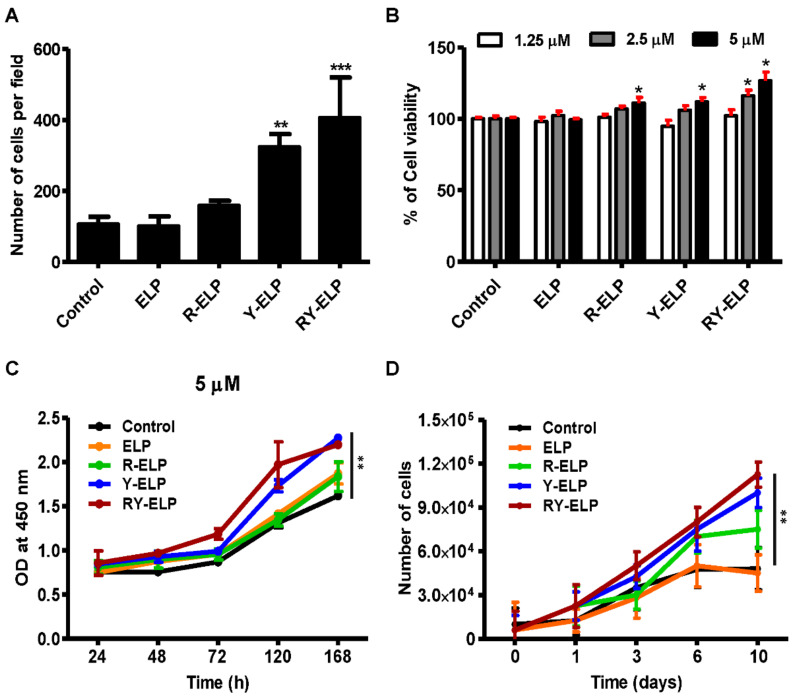
Cell viability and proliferation assay of MSCs. (**A**) Quantification of adhered cells observed per field (*n* = 3). ** *p* < 0.01 and *** *p* < 0.001 significant difference of Y-ELP or RY-ELP compared with uncoated control. (**B**) Cell viability of MSCs was evaluated post 24 h using CCK8 (*n* = 3). * *p* < 0.05 significant difference of fusion ELPs compared with uncoated surface. (**C**) Relative cell proliferation rates at 1, 3, 7, and 10 days were determined by CCK-8 assay (*n* = 3). ** *p* < 0.01 significant difference RY-ELP compared with uncoated surface. (**D**) A total of 0.6 × 10^4^ cells of MSCs (at P3) were seeded on 48-well plates coated with ELP variants. The effect of fusion ELP on the proliferation of MSCs was monitored by measuring the cell population at different time intervals. ** *p* < 0.01 significance of RY-ELP compared with untreated control.

**Figure 6 biomedicines-10-01151-f006:**
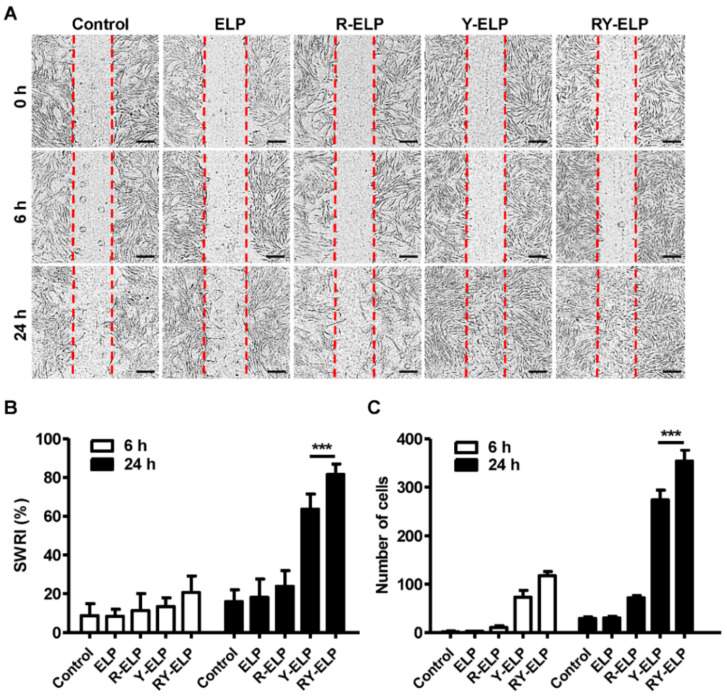
Migration assay. (**A**) Representative phase-contrast images of wounded MSCs cultured on a fusion-ELP-coated surface as compared with uncoated over a 24-h period (*n* = 3), Scale bars: 100 µm. (**B**) The bar graph represents the percentage of scratch wound recovery index (SWRI) at different time points. *** *p* < 0.001 significance difference of Y-ELP or RY-ELP compared with untreated control (*n* = 3). (**C**) Quantitative determination of total migratory MSCs at different time points (*n* = 3), *** *p* < 0.001 significance of Y-ELP or RY-ELP compared with untreated control.

**Figure 7 biomedicines-10-01151-f007:**
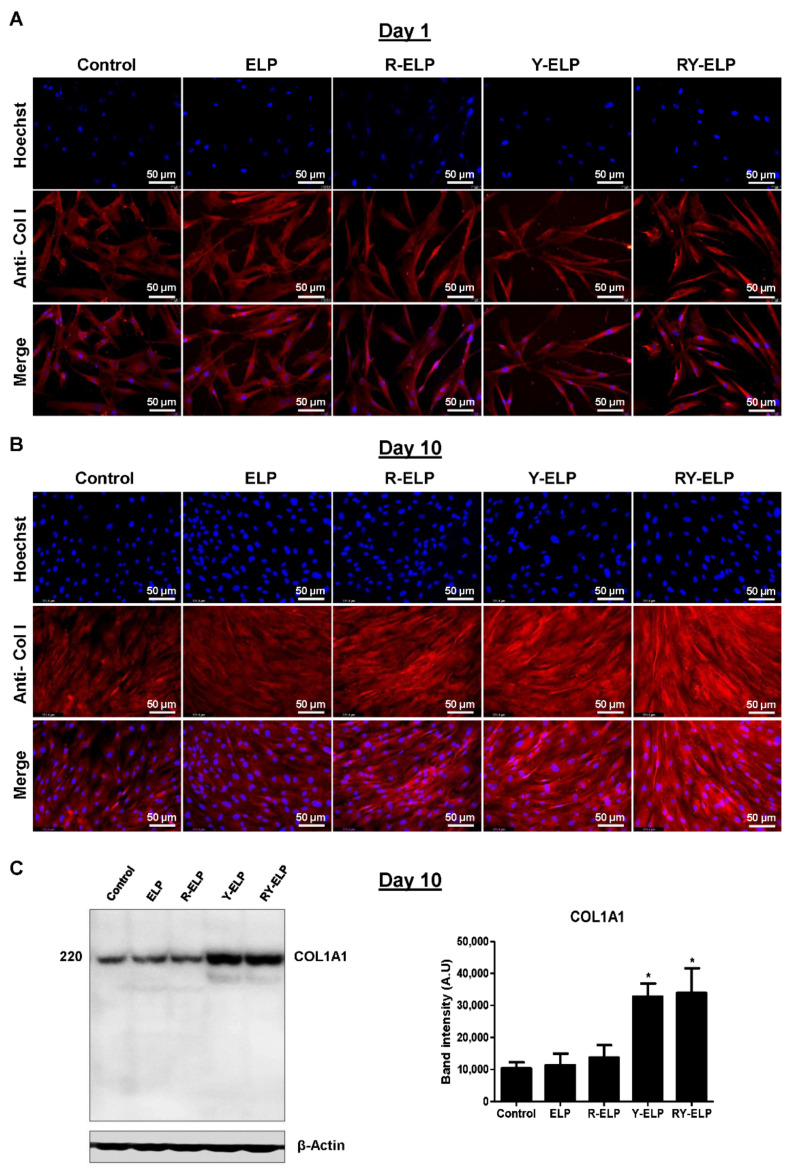
Determination of ECM secretion. Representative fluorescence images showing the expression of extracellular matrix proteins (type I collagen) using immunofluorescent staining of the MSC cultured for 1 day (**A**) or 10 days (**B**) on fusion-ELP-coated surfaces (*n* = 3), Scale bars: 50 µm. (**C**) Western blot showing the endogenous protein expression of (type I collagen) of MSCs cultured at day 10. The graph represents the quantification of Western band using image-J software. and data were expressed as mean ± SEM (*n* = 3), * *p*-value < 0.05.

**Figure 8 biomedicines-10-01151-f008:**
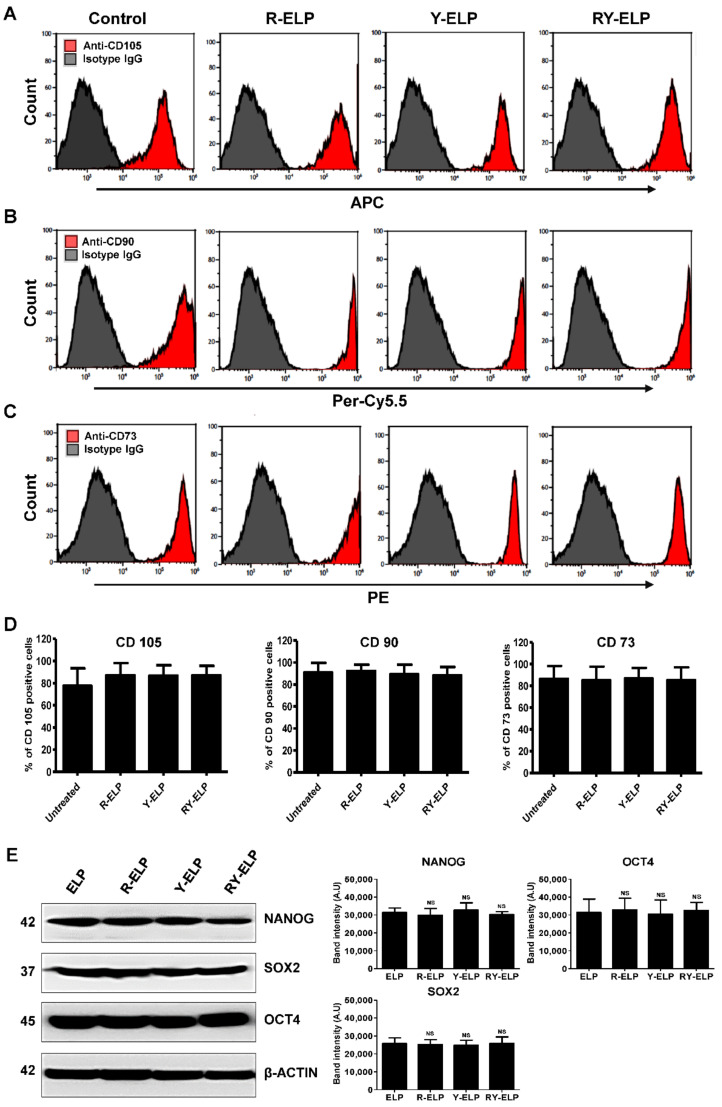
Evaluation of stemness properties of MSCs on ELP matrix. (**A**–**C**) Human bone-marrow-derived MSCs were analyzed for three positive mesenchymal stem cell markers CD73, CD90, and CD105 by flow cytometry. (**D**) The level of expression of mesenchymal stem cell markers observed in comparison with noncoated groups. The graph represents the mean ± SD (*n* = 3). (**E**) Western blot showing the endogenous protein expression of Nanog, SOX2, and OCT4 of MSCs cultured in coated plates and data were expressed as mean ± SD (*n* = 3), NS—not significant.

**Figure 9 biomedicines-10-01151-f009:**
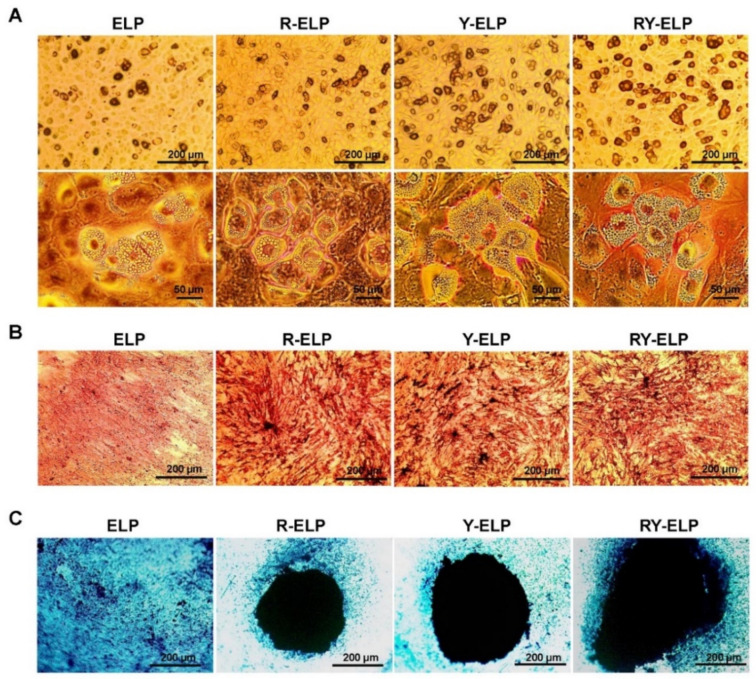
Trilineage differentiation in ELP matrix. (**A**) Adipogenesis was confirmed by cytochemical staining with Oil Red O to visualize lipid droplet formation (*n* = 3). (**B**) Osteogenesis was confirmed by cytochemical staining with Alzarin Red to visualize calcium nodule formation on fusion-ELP-coated surface compared with uncoated surface (*n* = 3). (**C**) Chondrocyte differentiation of MSCs on fusion-ELP-coated plates were visualized by Alcian Blue staining compared with uncoated control (*n* = 3), Scale bars: 200 µm.

## Data Availability

Not applicable.

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
