# Peer review of "Application of Bio-Active Elastin-like Polypeptide on Regulation of Human Mesenchymal Stem Cell Behavior"

_biomedicines, 2022, doi:10.3390/biomedicines10051151_

Round 1

Reviewer 1 Report

Why authors used directly one ml of bone marrow cells in culture. There should be a cell isolation step. Authors should have done ficoll density, positive selection or negative selection method etc. bone marrow cells come with a tremendous amount of junk containing immature cells including red blood cells, platelets macrophage etc. They should have shorted the cells using CD34 or other stem cell marker for MSCs progenitor cells.

ELP, R-ELP, Y-ELP and RY-ELP comes from BL-21 cells which could be tinted with lots of endotoxin. Please rule out endotoxin levels in culture. Please include the LAL test result in supplementary. Please rule the possibility of presence of endotoxin by other valid tests if you can't perform LAL. Please report IU of endotoxin.

Since ELP, R-ELP, Y-ELP and RY-ELP have no tags, detailed purification techniques need to be mentioned. Please describe how you have purified using Column after column or any enzymatic cleavages. Any available CD spectra for the structure?

Figure 3 cell migration assay: did authors have used Mitomycin for their experiments? If not, what is the rationale for not using mitomycin?

Authors need to put RT PCR for stemness candidates  grown in ELP, R-ELP, Y-ELP and RY-ELP plates and/or IF/FACS for nanong , OCT4, SOX2 or MYC. 

Author Response

Thank you for the valuable and critical comments. Your comments has helped us to tremendously improve our manuscript.

Reviewer #1 has asked for some major and minor correction in our manuscript. Reviewer 1 had asked to perform RT PCR for stemness candidates grown in ELP, R-ELP, Y-ELP and RY-ELP plates and/or IF/FACS for nanog, OCT4, SOX2 or MYC. Accordingly, the endogenous protein expression of NANOG, OCT4 and SOX2 in treated BM-MSCs was measured by western blot in order to determine whether ELP, R-ELP, Y-ELP and RY-ELP effects stemness properties in treated BM-MSCs, no significant changes in protein expression of stemness candidates were observed.  These results were included in the revised manuscript as (Figure 8 E). Apart from this, we have tried to sincerely answer each and every question raised by the reviewer 1 and all the necessary changes were made in the manuscript as per suggestion.

Reviewer 2 Report

Peer-Review Biomedicines – 1582160

The manuscript entitled “Application of bio-active ELP on regulation of human mesenchymal stem cell behaviour” by Vijaya Sarangthem et al. constitutes a well-design study supporting the use of human elastin-like polypeptide-based ECM-mimic biopolymers to enhance human bone marrow-derived stem cells adhesion, proliferation and migration envisaging potential application on cell therapy strategies. The manuscript is well written, properly organized and fits within the scope of the journal Biomedicines (ISSN 2073-4360). However, some issues must be addressed before its consideration for publication.

Major issues:

  1. The authors should clarify clearly the novelty of this manuscript/strategy in the abstract or in the final part of the Introduction section.

  1. The title should contain the term “elastin-like polypeptide” instead of the abbreviation.

  1. Figure 2B and Figure 5B: Are the error bars from these graphs very small or absent? Also, some are in red, others not. The authors should explain their reasons and improve these 2 graphs.

  1. Figure 7: The authors should provide a quantitative analysis of type I collagen expression in the different conditions. Additional analysis such as RT-qPCR or western blot analysis will also increase the quality of the study.

  1. Subsection 3.9: The values included in the manuscript text must include SD and the authors should provide a table (e.g., Supporting Material) with the values ± SD for the CD73, CD90 and CD105 expressions in the different experimental conditions.

  1. A major limitation of this manuscript is the lack of comparison/relation/discussion of the results obtained with other previously published studies (Results and Discussion section). The authors must address this issue in the revision process.

Minor issues:

  1. (Page 1, line 18): In the 1st phrase of the Abstract the term “unbeatable” strategies seems not appropriate. It is better to replace it for example by “promising” strategies.
  2. (Page 1, line 21): Please remove the misplaced word “cellular” to obtain “(...) stem cell’s function (...)”.
  3. (Page 1, line 35): Please remove the terms “in the best way” to obtain “(...) stimulatory response to support stem cell proliferation.”
  4. Throughout the all manuscript (specified in lines 40 and 305) the authors should define MSCs as mesenchymal stem/stromal cells. https://stemcellsjournals.onlinelibrary.wiley.com/doi/full/10.1002/sctm.17-0051
  5. (Page 2, line 90): It is more appropriate to replace “Relatively” by “Accordingly”.
  6. (Page 2, line 100): Please replace “maintaining” by “maintenance of”.
  7. (Page 3, line 109): Please add missing final point “.” -> “(...) CO2. Bone marrow (...)”
  8. (Page 3, subsection 2.3., line 137): Please specify the fluorescence microscopy equipment/brand.
  9. (Page 3, subsection 2.3.): What was the rationale to use a concentration of 5μM of fusion-ELPs?
  10. (Page 3, line 146): Please correct the phrase by adding the words “the cells were” to obtain: “(...) with PBS and the cells were stained (...)”
  11. (Page 4, lines 167/168): Please remove the repeated sentence.
  12. (Page 4, line 187): Please remove the misplaced “was” to obtain “(...) 10% FBS supplemented with (...)”
  13. (Page 5, subsection 2.10., line 205): Please specify the concentration of Oil Red O staining solution used.
  14. Throughout the Materials and Methods section, standardize the use of “ddH2O” and doubled distilled H2
  15. (Page 5, subsection 2.13.): Please specify the standard number of experiments/samples considered and the post hoc test used in the statistical analysis. Also specify all the different P-values (*P, **P, ***P).
  16. (Page 5, line 246): The segment “mainly found in many tissues” is not correct. Please remove “mainly” or specify the main tissues in which Elastin is present.
  17. (Page 7, line 309): Please correct the verb form to “were”: “(...) Fibroblasts were (...)”
  18. (Page 7, lines 312/313): Please correct “proliferative” to “proliferation”.
  19. (Page 9, lines 366): Please correct “in” to “of”: “(...) on cellular adhesion of human bone marrow-derived mesenchymal stem cells (BM-MSCs)”.
  20. (Page 10, lines 375): The text segment “well-attachment morphology” is not correct and must be rephrased.
  21. (Fig. S2 caption): It is better to replase “ELISA” by “plate” reader.
  22. (Page 11, line 396): Please add “in” to obtain: “(...) cell index ratio in comparison to (...)”.
  23. (Page 11, line 410): Please erase “growth and” or “and proliferation” since its just a repetition of the same concept.
  24. (Page 11, line 415): In the context of this subsection, it does not make sense to mention “and differentiation”. It should be removed.
  25. Throughout the manuscript (lines 457, 459, 463, 488, 512, 515, 517, 566,...) the authors often use BM-MSCs stem cells or BM-MSCs cells, which is repetitive. It is better to use just BM-MSCs.
  26. (Page 15, line 498): Please correct “BM-MScs” to “BM-MSCs”.
  27. (Page 17, line 574): Please correct “dysfunction” to “dysfunctional” to obtain “(...) repairing dysfunctional tissues (...)”

Author Response

Thank you for your valuable and critical comment. Your comments have help us to tremendously improve our manuscript.

Reviewer #2 had raised few major and minor queries. Reviewer #2 asked to provide a quantitative analysis of Type I collagen expression in the different conditions by rt-PCR or western blot. Accordingly, endogenous protein expression level of Type I Collagen in BM-MSCs grown in ELP, R-ELP, Y-ELP and RY-ELP coated plates were checked after day 10. The results of Western blot analysis of BM-MSCs grown on ELP, R-ELP, Y-ELP, and RY-ELP coated plates indicated that Y-ELP and RY-ELP coated plates significantly increased the amount of Type I collagen protein expression compared to controls and ELP coated plates. This result has been included in the revised manuscript as (Figure. 7 C). In addition, we have addressed all the query and minor revision as per suggestion.

Round 2

Reviewer 1 Report

Bone marrow cells typically need to be 10 times diluted for ficoll density. I don't buy that logic..you can use magnetic beads. Please avoid glorifying heterogeneous plating.

  Overall thank you for your answers. 

Author Response

Comments and Suggestions for Authors

Reviewer: Bone marrow cells typically need to be 10 times diluted for ficoll density. I don't buy that logic.. you can use magnetic beads. Please avoid glorifying heterogeneous plating.

Answer: Thank you for your valuable suggestion. In future we will further standardized Bone marrow cells harvesting and culturing method and try to avoid heterogenous plating. Thank you once again for minute details which help us in revising our manuscript and making it more meaningful research article.

Reviewer 2 Report

The manuscript entitled “Application of bio-active Elastin-like polypetide on regulation of human mesenchymal stem cell behaviour” by Vijaya Sarangthem et al. was considerably improved by the authors after this round of revisions. Therefore, I recommend that the manuscript should be accepted for publication after few minor corrections.

  1. Some error bars from graphs in Figure 2B and Figure 5B are still not identified/highlighted in red. For consistency, the authors should mark all of them.
  2. In the revised version there are still some occurrences of "BM-MSCs cell" (lines 28, 241, 608-subsection 3.10 title, 672 and 686) that should be corrected.

Author Response

Thank you for the critical comments. As suggested the error bar has been made visible in red color in Fig 2B and Figure 5 B. Please find the changes in the revised manuscript.

Occurrences of "BM-MSCs cells has been rectified in the entire manuscript. 

Thank you once again for the minute details and making the research article more meaningful.
